# The GOLIATH Project: Towards an Internationally Harmonised Approach for Testing Metabolism Disrupting Compounds

**DOI:** 10.3390/ijms21103480

**Published:** 2020-05-14

**Authors:** Juliette Legler, Daniel Zalko, Fabien Jourdan, Miriam Jacobs, Bernard Fromenty, Patrick Balaguer, William Bourguet, Vesna Munic Kos, Angel Nadal, Claire Beausoleil, Susana Cristobal, Sylvie Remy, Sibylle Ermler, Luigi Margiotta-Casaluci, Julian L. Griffin, Bruce Blumberg, Christophe Chesné, Sebastian Hoffmann, Patrik L. Andersson, Jorke H. Kamstra

**Affiliations:** 1Institute for Risk Assessment Sciences, Department of Population Health Sciences, Faculty of Veterinary Medicine, Utrecht University, 3508 TD Utrecht, The Netherlands; j.h.kamstra@uu.nl; 2INRAE Toxalim (Research Centre in Food Toxicology), Metabolism and Xenobiotics (MeX) Team, Université de Toulouse, INRAE, ENVT, INP-Purpan, UPS, 31027 Toulouse, France; daniel.zalko@inrae.fr (D.Z.); fabien.jourdan@inrae.fr (F.J.); 3Centre for Radiation, Chemical and Environmental Hazards, Public Health England, Chilton OXON. OX11 0RQ, UK; Miriam.Jacobs@phe.gov.uk; 4Institut NUMECAN (Nutrition Metabolisms and Cancer) INSERM UMR_A 1341, UMR_S 1241, Université de Rennes, F-35000 Rennes, France; bernard.fromenty@inserm.fr; 5Institut de Recherche en Cancérologie de Montpellier (IRCM), INSERM U1194, ICM, Université de Montpellier, 34298 Montpellier, France; patrick.balaguer@inserm.fr; 6Center for Structural Biochemistry (CBS), INSERM, CNRS, Université de Montpellier, 34090 Montpellier, France; william.bourguet@cbs.cnrs.fr; 7Department of Physiology and Pharmacology, Karolinska Institutet, 17177 Stockholm, Sweden; vesna.munic.kos@ki.se; 8IDiBE and CIBERDEM, Universitas Miguel Hernandez, 03202 Elche (Alicante), Spain; nadal@umh.es; 9ANSES, Direction de l’Evaluation des Risques, Agence Nationale de Sécurité Sanitaire de l’Alimentation, de l’Environnement et du Travail, 14 rue Pierre et Marie Curie, 94701 Maisons-Alfort CEDEX, France; claire.beausoleil@anses.fr; 10Department of Biomedical and Clinical Sciences (BKV), Cell Biology, Medical Faculty, Linköping University, SE-581 85 Linköping, Sweden; susana.cristobal@liu.se; 11Sustainable Health, Flemish Institute for Technological Research, VITO, 2400 Mol, Belgium; sylvie.remy@vito.be; 12Department of Life Sciences, College of Health and Life Sciences, Brunel University London, Uxbridge UB8 3PH, UK; Sibylle.Ermler@brunel.ac.uk (S.E.); Luigi.Margiotta-Casaluci@brunel.ac.uk (L.M.-C.); 13Section of Biomolecular Medicine, Division of Systems Medicine, Department of Metabolism, Digestion and Reproduction, Imperial College London, South Kensington, London SW7 2AZ, UK; jlg30@ic.ac.uk; 14Department of Developmental and Cell Biology, University of California Irvine, 2011 BioSci 3, University of California, Irvine, CA 92697-2300, USA; blumberg@uci.edu; 15Biopredic International, Parc d’Activité de la Bretèche Bâtiment A4, 35760 Saint Grégoire, France; christophe.chesne@biopredic.com; 16Seh Consulting + Services, 33106 Paderborn, Germany; sebastian.hoffmann@seh-cs.com; 17Chemistry Department, Umeå University, SE-90187 Umea, Sweden; patrik.andersson@umu.se

**Keywords:** risk assessment, chemicals, endocrine, obesity, diabetes

## Abstract

The purpose of this project report is to introduce the European “GOLIATH” project, a new research project which addresses one of the most urgent regulatory needs in the testing of endocrine-disrupting chemicals (EDCs), namely the lack of methods for testing EDCs that disrupt metabolism and metabolic functions. These chemicals collectively referred to as “metabolism disrupting compounds” (MDCs) are natural and anthropogenic chemicals that can promote metabolic changes that can ultimately result in obesity, diabetes, and/or fatty liver in humans. This project report introduces the main approaches of the project and provides a focused review of the evidence of metabolic disruption for selected EDCs. GOLIATH will generate the world’s first integrated approach to testing and assessment (IATA) specifically tailored to MDCs. GOLIATH will focus on the main cellular targets of metabolic disruption—hepatocytes, pancreatic endocrine cells, myocytes and adipocytes—and using an adverse outcome pathway (AOP) framework will provide key information on MDC-related mode of action by incorporating multi-omic analyses and translating results from in silico, in vitro, and in vivo models and assays to adverse metabolic health outcomes in humans at real-life exposures. Given the importance of international acceptance of the developed test methods for regulatory use, GOLIATH will link with ongoing initiatives of the Organisation for Economic Development (OECD) for test method (pre-)validation, IATA, and AOP development.

## 1. Introduction

The incidence of obesity and metabolic disorders such as diabetes and non-alcoholic fatty liver disease (NAFLD) has reached “Goliathan” proportions. In Europe alone, more than 50 million people suffer from one or a combination of these disorders [1,2]. The worldwide increase in metabolic disorders cannot be explained by lifestyle and genetic factors alone; the role of environmental factors in these disorders has been increasingly acknowledged [3]. Exposure to endocrine-disrupting chemicals (EDCs) that disrupt metabolic functions—chemicals collectively referred to as “metabolism disrupting chemicals” (MDCs)—is an environmental risk factor that urgently requires more attention. MDCs are natural and anthropogenic chemicals that can promote metabolic changes, which can ultimately result in obesity, diabetes, and/or fatty liver in humans [4]. Given the important role these metabolic alterations can play in the global epidemics of metabolic disorders, international chemical regulations must pay due attention to the identification of MDCs and the assessment of the risk associated with exposure.

Within European chemicals regulations, criteria to identify EDCs have been recently proposed that require information on a chemical’s endocrine mode of action (MoA) and related adverse effects relevant for human health [5]. This involves the screening and testing of EDCs according to the EU Test Methods Regulation [6], which mainly incorporates internationally accepted test methods developed under the Organisation for Economic Cooperation and Development (OECD) [7]. The current validated and standardized test methods to identify EDCs are principally focused on well-studied endocrine pathways in the estrogen, androgen, and thyroid systems. However, there are no suitable in vivo or in vitro tests for regulatory purposes that identify the potential metabolic and metabolism disrupting effects of chemicals. The need for these tests has been internationally recognized (e.g., [8,9,10,11]), as, without them, comprehensive hazard and risk assessment of chemicals for potential metabolic disrupting activity is virtually impossible.

In January 2019, the five-year European research project “Beating Goliath: Generation of Novel, Integrated and Internationally Harmonized Approaches for Testing Metabolism Disrupting Chemicals” (GOLIATH) started, with a consortium comprised of world-leading experts in endocrinology and molecular biology, endocrine disruption, toxicology, epidemiology, bioinformatics, test method development, (pre-)validation, and chemical regulation. The overall aim of the GOLIATH project is to improve hazard assessment of MDCs by generating novel, optimized, integrated, and internationally harmonized approaches for testing metabolic disruption. The GOLIATH project spans the entire spectrum of testing, from in silico predictive modeling and high throughput screening, to the development of robust ready-to-use in vitro assays and improving the utility of current in vivo testing guidelines. By incorporating novel omics technologies, and translating in vitro and in vivo assays to human health effects, GOLIATH will generate new integrated approaches to testing and assessment for MDCs, incorporating mainly in vitro test methods, as well as providing novel insights in the mechanisms by which MDCs disrupt metabolic pathways and induce adverse effects on human health. GOLIATH is one of eight projects awarded funding within EURION, the “European *cluster* to improve identification of endocrine disruptors” (see also https://eurion-cluster.eu/).

This project report has the following main objectives (1) to introduce the background to the project, i.e., metabolic disorders and the major classes of MDCs that will be studied in the GOLIATH; (2) to describe the approaches developed in the GOLIATH project; (3) to provide a brief state-of-the-art review of the approaches that have been used up to now to assesses metabolic disrupting activity and related adverse effects of EDCs. A final objective of this project report is to raise awareness of the scientific community of the forthcoming data that will be generated in this project, ensuring that the data will be open access, in line with the “Findable, Accessible, Interoperable, Reusable (FAIR) initiative [12].

## 2. Metabolic Disorders

GOLIATH focuses on developing test methods that will contribute to the assessment of the role of MDCs in obesity and metabolic disorders including insulin resistance (IR), type 2 diabetes (T2D), and NAFLD (Figure 1). The WHO defines obesity as “abnormal or excessive fat accumulation that may impair health,” and diabetes as a “chronic disease caused by inherited and/or acquired deficiency in production of insulin by the pancreas, or by the ineffectiveness of the insulin produced” [1]. NAFLD is a disease “characterized by excessive hepatic fat accumulation, defined by the presence of steatosis in >5% of hepatocytes” [13]. Maintenance of plasma glucose levels within the physiological range is based on a negative feedback system between insulin production and release by pancreatic β-cells and insulin response by insulin-sensitive tissues, mainly liver, adipocytes, and skeletal muscle. IR is a hallmark of obesity, NAFLD, and a predecessor of T2D, and can develop in response to environmental factors, such as aging, obesity, and exposure to MDCs. In susceptible individuals, environmentally-induced peripheral IR raises blood glucose levels, which, in turn, stimulate insulin secretion by pancreatic β-cells. The resulting hyperinsulinemia will cause further IR, which may generate a vicious circle that leads to β-cell failure, reduced β-cell mass, and, ultimately, T2D (Figure 1).

### Metabolic Disrupting Chemicals

New discoveries made in the last 15 years have provided solid evidence that MDCs cause IR in peripheral tissue and alter β-cell mass and function [4,14,15]. Furthermore, MDCs can act as “obesogens,” inducing adipogenesis as well as hyperplasia and hypertrophy of adipocytes in white adipose tissue (WAT) [4]. Evidence suggesting the effects of MDCs on other forms of adipocytes, such as thermogenic beige/brite and brown fat cells, is emerging but is still an understudied area of research. MDCs can also directly affect the liver, promoting IR and de novo lipogenesis, thus favoring fatty liver disease development and progression [4]. Development of fatty liver is a major health issue since this lesion can progress to non-alcoholic steatohepatitis (NASH) and then to severe liver diseases including cirrhosis and hepatocellular carcinoma [2]. Studies in vivo indicate that skeletal muscle might be a target tissue of MDCs as well, although evidence of a direct effect on isolated cells is still very scarce. MDC exposure is thus a risk factor for obesity, and IR, and increases the risk of developing T2D, NAFLD, and other metabolism-related diseases.

The list of chemicals implicated in metabolic disorders is growing and includes bisphenols, pesticides, phthalates, metals, and perfluorinated compounds [4]. Recent literature reviews are available that synthesize the reported metabolism disrupting effects of these chemicals in laboratory studies [4,15,16]. For example, bisphenol A (BPA) has been shown to directly stimulate β-cells and cause primary hyperinsulinemia after amplification of glucose-stimulated insulin secretion in rodents, leading to IR in skeletal muscle and liver [17]. BPA is also known to induce steatosis in human hepatocytes [18] and has obesogenic properties, whereas developmental BPA exposures in rodents are associated with increased adiposity [19]. A brief overview of the initial set of MDCs that will be tested in GOLIATH is shown in Table 1. This initial set of model test chemicals was selected because they have a wide range of mechanisms of action (MoA), and there is existing data from animal and/or human studies that indicate metabolism disrupting effects following exposure. In addition, all of these six chemicals have been detected in humans in biomonitoring studies. A summary of human epidemiological studies linking (early life) exposure and metabolic disorders in these six chemicals from the literature are shown in Appendix A. An overview of the available human biomonitoring data for four of these chemicals in Belgian cohorts is given in Appendix A, while for tributyltin (TBT) and diphenyl phosphate (DHPH, a metabolite of triphenyl phosphate (TPP)), an overview of international biomonitoring data is provided in Appendix A.

## 3. Test Methods in GOLIATH to Determine Metabolism Disrupting Activity

The integrated structure of GOLIATH is straightforward (Figure 2): using an Adverse Outcome Pathway (AOP) framework, we will develop test methods that cover the pathways from the molecular initiating event (MIE) to a key event in cells and tissues, to the adverse metabolic outcome in humans. To this end, for MIE testing, we will develop in silico predictive models based on MIEs, molecular dynamics to predict human nuclear receptor (NR) binding; high throughput screening (HTS) assays to confirm NR binding and transactivation. Specific in vitro models will be developed using key target tissues (adipose, liver, endocrine pancreas, skeletal muscle) to model key cellular events that may lead to adverse outcomes. We will further develop an alternative zebrafish assay for metabolic disruption and use this assay, together with data from existing in vivo rodent assays to predict adverse outcomes in humans. We will work closely with epidemiologists to ensure the translation of lower and higher-tiered methods to the human situation. In the following section, we will briefly introduce each of these models, as well as review relevant literature available on testing MDCs in these models.

### 3.1. Test Methods to Determine MIEs

#### 3.1.1. In Silico Predictive Models

In the first three years of the project, GOLIATH will develop and apply alternative methods, carried out solely in a computer-based interface to obtain information on the hazard of a substance. The most prominent in silico approaches for the generation of predictive data are (Quantitative) Structure-Activity Relationships ((Q)SARs), category formation, and read-across methods. In silico screening methods have been proposed for numerous endpoints (e.g., profilers for DNA binding in QSAR Toolbox [20]) and their use has been encouraged from regulatory bodies to support decision making (e.g., REACH legislation (2007)) with guidance on validation principles [21]. The main drive and limitation for the development of reliable computational approaches is high-quality data. These data together with the applied methodology define the applicability domain of the model which is a key concept of in silico modeling, i.e., the defined chemical space for which a computational method can reliably provide predictions.

As reviewed in Schneider et al. [22], the majority of in silico studies for identifying potential MDCs and studying NR mediated interactions involved in metabolic disorder apply (Q)SAR models. Further, physiologically-based toxicokinetic/pharmacokinetic (PBTK/PBPK) models have been used for prediction of absorption, distribution, metabolism, and excretion (ADME) processes and facilitation of quantitative in vitro-in vivo extrapolation (QIVIVE). Notable examples of MDC specific PBTK models have been suggested for BPA [23], and TPP [24]. (Q)SAR models have been developed for the NRs involved in metabolism disrupting effects, i.e., PPARα, PPARγ, PXR, CAR, LXRα, LXRβ, FXR, for endpoints such as median effective concentration (EC50), and inhibitory affinity constant (K_i_) for defined applicability domains (Appendix A). These include local models for CAR activity with a limited set of chemicals (e.g., [25]) to large initiatives using Tox21 HTS data for FXR [26] and methodologies spanning from combining molecular docking and pharmacophore filtering to identify agonists of PPARγ [27] to classification models identifying chemicals activating and antagonizing PXR [28].

The growing number of 3D crystal structures of protein targets in complex with ligands provides a wealth of structural information that can be used to predict their interactions with EDCs using docking and scoring procedures. GOLIATH uses and further develops the online prediction tool called EDMon (Endocrine Disruptor Monitoring) dedicated to ligand screening of nuclear receptors which relies upon known crystal structures or models and the availability of large experimental ligand affinity datasets. The server is currently available (http://edmon.cbs.cnrs.fr) to screen for estrogen receptors (ERα/β) and the peroxisome proliferator-activated receptor γ (PPARγ). By uploading compound information in an appropriate (mol2) format, the user can get predictions of the binding modes and affinities using a rescoring approach based on machine learning [29]. Within GOLIATH, this tool will be extended to include NRs identified as key MIEs in metabolism disruption.

#### 3.1.2. NR Screening

The GOLIATH project will use reporter cell lines stably expressing the ligand-binding domain (LBD) of human PPARα, PPARγ, LXRα, LXRβ, PXR, or CAR fused to the yeast GAL4 DNA binding domain (DBD) and the luciferase under the control of five GAL4 responsive elements to screen the initial set of GOLIATH compounds (Table 1) as well as a large number of environmental compounds. These reporter cell lines were generated by a two-step transfection procedure [30]. First, a stable cell line, HG5LN, expressing only the reporter gene was developed. These cells were then transfected with the different receptor genes. These six nuclear receptors reporter cell lines are powerful tools to characterize the nuclear receptor activity of EDCs in a standardized, high-throughput screening technique. In addition, the HG5LN parental cell line is used as a negative control. These cellular assays have previously shown that BPA activates PXR but no PPARγ while TBBPA activates both receptors [31,32]. In the first two years of the project, a library of hundreds of candidate MDCs will be screened with these reporter gene assays to identify mechanisms of action. In addition, up to two of these reporter gene assays will be further developed and optimized as candidate assays to go forward to pre-validation.

#### 3.1.3. Bioactive Thermal Protein Profiling (bTPP)

GOLIATH will apply the bioactive thermal proteome profiling (bTPP) methodology to provide a comprehensive assessment of cellular proteins that interact with MDCs. This assessment includes the identification of the protein targets, determination of the differential affinity among targets, and prediction of a network of molecular pathways and the mechanisms of actions [33]. The evaluation of the parameters that define the target engagement has been traditionally performed in single-parameter assays until the development of the cellular thermal shift assay (CETSA) [34]. This method was first used to measure the increase of thermal stability to the unfolding of proteins that were interacting with other compounds. Thermal proteome profiling is a subsequent development of the CETSA that uses quantitative mass spectrometry to identify and quantity the folded and unfolded proteins from a proteome after a thermal shift assay. This method has been successfully applied to study drug targets and off-targets [35,36], as well as for deciphering drug–target interaction [37,38] and defining specific mechanisms of action in very complex samples [39]. In the GOLIATH project, we will spend the first three to four years developing a modified method of thermal protein profiling, called bTPP [33] to identify MIEs in cells and zebrafish.

### 3.2. Test Methods to Determine Key Events, i.e., In Vitro Models of Metabolic Disruption

As mentioned above, in vitro models of metabolic disruption will be developed for the liver, pancreas, skeletal muscle, and adipose tissue, for which the methods to be developed are outlined below. We acknowledge that metabolism involves a complex interplay between multiple organs and tissues in the body, including inter alia the brain and the gut, but decided to focus on liver, pancreas, skeletal muscle, and adipose tissue, which we consider as key organs for metabolic disruption. However, we will work together with other projects in the EURION cluster that investigate those additional organs and tissues.’

#### 3.2.1. Liver

Investigations regarding MDC metabolism and metabolic disruption in hepatocytes are carried out in HPR116 cells (Biopredic International, Saint-Grégoire, France), which are cryopreserved differentiated HepaRG cells. The HepaRG cell line is derived from a liver tumor of a female patient suffering from hepatocarcinoma [40]. Notably, differentiated HepaRG cells express xenobiotic-metabolizing enzyme (XME) activities close to those measured in primary human hepatocyte cultures [41,42]. Furthermore, these cells also express several key transcription factors regulating XME expression such as AhR, PXR, CAR, and PPARs [42,43,44]. Of note, the differentiated HepaRG cells present mitochondrial function close to primary human hepatocytes [45,46]. More generally, recent investigations reported that the deep proteome of HepaRG cells is overall similar to that of primary human hepatocytes [47].

##### Metabolism in HPR116 Cells

The metabolism of xenobiotics and endogenous metabolism are closely intertwined as they rely on a common pool of enzymes, regulated by specific NRs. For this reason, the modulation (induction, inhibition) of XME activities by xenobiotics can have a major impact not only on their own metabolic fate but also on endogenous anabolic and catabolic metabolic pathways. In the case of MDCs, these are potential mechanisms of action that need to be better investigated. In GOLIATH, the functional metabolic capabilities of HPR116 cells will be assessed using a battery of phase I and phase II XME functional activities measurement assays, with both short term and long term exposure. For phase I reactions, these include the characterization of cytochrome P450 (CYP) activities using [^14^C]-labeled testosterone. The formation of the different metabolites of testosterone through CYP450 oxidations is regio-selective in humans [48], thus allowing to estimate the functional activities of major CYP isoforms based on the quantification of specific testosterone metabolites. For phase II reactions, the characterization of UDP-glucuronyl transferases (UGTs) and sulfotransferases (SULTs) activities will be achieved using probe substrates such as 4-methyl-umbelliferone and 7-hydroxycoumarin.

In addition, extensive work will be carried out in the first two years of GOLIATH to expand the chemical applicability domain of the “CYP induction assay.” This assay was originally validated with pharmaceuticals within the EURL ECVAM Multi-study Validation Trial [49] and was recently summarized and published by Bernasconi et al. [50]. The CYP induction assay aims at assessing the potential of xenobiotics to induce key CYP activities in vitro (CYP1A, CYP2B, and CYP3A). CYP1A2, CYP2B6, and CYP3A4 are globally accepted as biomarkers of CYP induction in the regulatory guidelines of pharmaceutical agencies [50]. CYP induction processes rely on the binding/activation of specific transcription factors and NRs such as AhR, CAR, and PXR which can constitute MIEs involved in the onset of putative adverse effects of MDCs. In GOLIATH, the objective is to support the validation (Bernasconi et al.) by extending the chemical applicability domain of the CYP induction assay, beyond the validation reference chemical set of pharmaceuticals, to achieve a better prediction of the capability of industrial chemicals and pesticides to induce human CYP activities. Using the existing standard operating procedure (SOP) detailed in an OECD draft test guideline [51], we will use HPR116 cells exposed to selected test chemicals for 48 h to determine key CYP activities following the use of a cocktail of specific probe substrates (phenacetin, bupropion, midazolam) and subsequent LC/MS-MS analyses of product metabolites. The expansion of the applicability domain of the CYP induction assay to industrial and pesticidal chemicals, including MDCs, will allow us to define the optimal use of this test method within approaches for testing strategies for metabolic disruption.

##### Steatosis

We will develop in vitro models of both steatosis (i.e., fat accumulation) and IR in HPR116 cells in the first three to four years of GOLIATH. Numerous investigations have been performed in HepaRG cells to study xenobiotic-induced toxicity and related mechanisms [42,52,53,54]. More specifically, this cellular model has been used to study steatosis induced by different drugs known to be steatogenic in humans, including amiodarone, tamoxifen, tetracycline, and valproic acid [55,56,57]. Furthermore, HepaRG cells are a relevant model to investigate different mechanisms leading to steatosis including inhibition of mitochondrial fatty acid oxidation and activation of de novo lipogenesis [55,57,58]. However, so far, we are aware of only two studies in HepaRG cells reporting the steatotic effect of BPA [18] and TBT [59]. In our previous work on BPA, steatogenic effects were demonstrated for a concentration as low as 2 nM, in proliferating and early differentiating HepaRG cells, allowing to mimic a putative perinatal exposure scenario [18]. However, the steatogenic effect of BPA in fully differentiated HepaRG cells has not yet been assessed. Studies on potential steatotic effects of other chemicals in the initial set of GOLIATH (e.g., PFOA, TPP, pp’-DDE, and TCS) have not yet been reported in the scientific literature.

#### 3.2.2. Endocrine Pancreas

Pancreatic β- and α-cells play an essential role in glycemic control through the secretion of insulin and glucagon. T2D is characterized by hypoinsulinemia and hyperglucagonemia with elevated blood glucose levels. Evidence indicates that MDCs alter the functional mass of β-cells, either increasing or decreasing glucose-stimulated insulin secretion (GSIS) as well as β cell division and death [60,61]. There are fewer examples of MDCs affecting α-cells but it is known that BPA rapidly alters Ca2+ signaling in response to glucose in ex vivo studies [62] and perinatal exposure decreases glucagon levels in offspring [63]. Therefore, in GOLIATH, we will dedicate four to five years to develop and characterize novel cellular testing methods to screen the capacity of MDCs to affect function and survival of pancreatic β cells (EndoC-βH1 and INS-1E) and α cells (α-TC1-9).

#### 3.2.3. Muscle

Skeletal muscle (SM) plays a crucial role in glucose homeostasis. Insulin binding to its receptors on the cell membrane of SM initiates a signaling cascade of phosphorylation eliciting the translocation of vesicles containing glucose transporter 4 (GLUT4) to the plasma membrane initiating glucose transport by facilitated diffusion. Insulin resistance is a primary defect in T2D. Despite evidence suggesting that MDCs induced insulin resistance in SM [17,64], there are studies concerning MDCs action on SM are scarce. In GOLIATH we will use the mouse myocyte cell line C2C12 to evaluate insulin signaling upon MDCs exposure, measuring the expression of proteins such as GLUT4 and protein kinase B/AKT.

#### 3.2.4. Adipocytes

##### Adipocyte Commitment and Differentiation

Within the first two to three years of the project, GOLIATH will focus on developing human-relevant in vitro models to test the effects of MDCs on the differentiation of white, beige/brite, and brown adipocytes. To date, most studies that focused on MDCs and enhanced adipocyte formation have utilized the murine 3T3-L1 pre-adipocyte cell line. With respect to the initial GOLIATH chemical selection, in 3T3-L1 cells, the following MDCs have tested positive for stimulating adipocyte differentiation at concentrations indicated in Table 1: BPA (10 nM–100 μM) [65,66,67,68,69,70], TBT (10–100 nM) [71,72,73,74,75], PFOA (10–100 μM) [71,76], pp’-DDE (30–100 μM) [77,78], and TPP (10–25 μM) [79]. In contrast, less research has been performed on more relevant human models. As the murine 3T3-L1 cell line is already committed to the adipocyte lineage, it provides limited information compared to multipotent human mesenchymal stem cells (hMSCs). As hMSCs remain multi-potent, they can be programmed to differentiate into a variety of cell types, including white, beige/brite, and brown adipocytes. In GOLIATH, we will use commercially available hMSCs to evaluate the effects of MDCs for their ability to: (1) differentiate MSCs into adipocytes (standard adipogenesis assay); (2) commit MSCs to the adipocyte lineage; (3) interfere with (or promote) generation of thermogenic beige/brite adipocytes. The commitment [80] and beiging [81] assays have been applied successfully in mouse bone marrow-derived MSCs following exposure to TBT and RXR activators and will be adapted in GOLIATH to the human model. The standard adipogenesis assay with hMSCs has been previously tested, showing an increase in adipogenesis with TBT (50 nM) [82] and PFOA (1 nM to 1 μM) [83]. TCS showed inhibition in adipogenesis (1.25 and 2.5 μM) [84], whereas BPA showed no effects up to 1 μM [85]. No literature is available on the adipogenic capacity of p,p’-DDE and TPP in hMSCs.

##### Insulin Resistance

In addition to developing methods to test adipocyte differentiation, we will develop methods for determining insulin resistance in adipocytes, to be completed by year 3. To our knowledge, there have been no reports on inducing insulin resistance in adipocytes differentiated from human mesenchymal stem cells. However, insulin resistance-like changes in adipocytes have so far been characterized by murine preadipocyte cells, 3T3-L1. Lo et al. [86] have described the effects of different insulin resistance inducing conditions, such as treatment with TNFα, dexamethasone, high insulin, and hypoxia in 3T3-L1 adipocytes, and compared them with primary murine adipocytes. The effects of MDCs on glucose consumption and insulin sensitivity of 3T3-L1 adipocytes are summarized in Table 2. Some MDCs have been shown to affect basal and insulin-stimulated glucose uptake, GLUT4, and adiponectin expression, as well as that of proinflammatory cytokines (Table 2). In GOLIATH we will use 3T3-L1 adipocytes and hMSCs differentiated into adipocytes to systematically analyze the effects of MDCs on glucose consumption and insulin sensitivity.

### 3.3. Methods and Approaches to Determine Adverse Outcomes

GOLIATH will determine the relevance of MDC exposure at the organism level by translating the impact of MDCs on metabolic function in vitro to the organism level, using zebrafish models, existing data from rodent studies, and studies in human cohorts.

#### 3.3.1. Zebrafish

A novel in vivo alternative test method to measure metabolic disruption in zebrafish will be developed within the first two years of the project. This test will be an extension of a recently developed method to measure the effects of MDCs on adipogenesis [94] and will use commercially available transgenic zebrafish lines that produce fluorescent pancreatic beta cells and measure effects on pancreas and liver function in addition to adipocyte size and number.

#### 3.3.2. Rodent Studies

GOLIATH will examine existing data generated from animal studies performed with the selected MDCs, and utilize this information to propose relevant tissues and endpoints where the current in vivo battery of test guidelines could be augmented. Note that though no new in vivo studies will be conducted within this project, partners within the GOLIATH project will team up with other partners within and outside the EURION cluster who are performing rodent studies to enable translational and cross-species comparative analyses.

#### 3.3.3. Epidemiological Studies

Within the framework of GOLIATH, we will explore epidemiologic associations in the “Flemish Environment and Health Studies” that have been running since 2002 (FLEHS, http://www.milieu-en-gezondheid.be/) [95,96,97,98,99,100,101,102] and in the “Prevention and Incidence of Asthma and Mite Allergy” cohort study (PIAMA, https://piama.iras.uu.nl/english/) that has been running since 1996 [103,104]. Both cohort studies contain a detailed characterization of lifestyle and environmental exposures as well as information on metabolically relevant molecular and physical outcome parameters at multiple ages in childhood and adolescence. As documented in Appendix A, exposure data on p,p’-DDE, PFOA, BPA, and TCS are available for different age groups within FLEHS. New PFOA data from sampled adolescents, is expected to become available within the coming year, as part of the HBM4EU project [105]. The PIAMA cohort will be enriched with untargeted high-resolution mass spectrometry data (exposome scan) to screen and quantify the MDCs and hundreds of other environmental chemicals in addition to endogenous metabolites [106]. We will investigate changes in anthropometric measures (BMI, waist circumference, waist-to-height ratio), concentrations of lipids (HDL-cholesterol, triglycerides), levels of leptin and insulin, and changes in -omics readouts in relation to the concentrations of the chemicals. The possible mediating effects of the molecular markers and pathways in the exposure–outcome associations will be explored. Findings will be compared with data collected from cellular, zebrafish, and (existing) in vivo models. Vice versa, putative predictive markers for screening of chemicals for metabolic effects identified within GOLIATH cellular, zebrafish and existing data from in vivo models will be examined in a subset of samples from the cohorts to confirm their human relevance and potential cross-species utility as biomarkers that could be used as part of the augmentation of in vivo test guidelines.

## 4. Approaches to Determine Endocrine Mode of Action

GOLIATH will provide critical information on the endocrine mode of action of chemicals by network modeling and multi-omic approaches. Using multi-omic (transcriptomics, metabolomics, and lipidomics) data from the in vitro models outlined above, metabolic MoAs will be defined as metabolic sub-networks (sets of potentially modulated metabolic reactions) by which MDCs exert metabolic disruption. Sub-network searches will be achieved using computational models of human genome-scale metabolic networks [107] and omics data obtained in cellular studies. The first step will consist in precising the endogenous metabolism of each in vitro model [108]. Then sub-networks will be extracted based on metabolomics and lipidomics data using network algorithms [109]. Resulting sub-networks will be stored in a newly created database (alongside the international repositories for each omic data, e.g., MetaboLights [110] for metabolomic data, ArrayExpress [111] for transcriptomics) and an online visual and interactive web interface will be developed to access these metabolic MoAs and to allow scientists to input their data to perform similar computations. This open platform will be built as an extension of the MetExplore [112] web server developed within the consortium.

## 5. From Individual Assays to AOPs and an Integrated Approach to Testing and Assessment of MDCs

Once standardized and optimized, in vitro test methods developed in GOLIATH will be defined and documented in a harmonized manner to facilitate a comparative assessment of the quality of data produced, and the potential utility in regulatory applications. Based on this information as well as mechanistic, strategic, and intellectual property considerations, sufficiently mature test methods will be prioritized for pre-validation, which will focus on test method transferability, reproducibility between laboratories, optimization of the protocol and preliminary predictivity. These activities will prepare the test methods for potential regulatory uptake. To facilitate this process, each test method will also be aligned with a network of relevant AOPs [113] that will be developed to synthesize the existing knowledge concerning the cascade of causally related key events that link the perturbation of specific molecular targets to the human adverse health outcomes considered within this project. These AOPs will provide a weight-of-evidence-based framework to support the selection of appropriate in vitro assays for the IATA, which will address the critical key events and key event relationships identified for metabolic disruption.

AOPs will also inform the development of an IATA for hazard identification, hazard characterization, and potential risk assessment of MDCs. The IATA will integrate and weigh all relevant evidence to inform regulatory decision-making regarding potential hazard and/or risk of MDCs. OECD IATA guidance will be utilized [114], and lessons learned from IATA projects in progress (e.g., Jacobs et al. [115]) will be applied. The IATA will be tested with selected chemicals to include them as part of the OECD case studies reporting cycle output. The objective of the IATA Case Studies Project is to increase experience with the use of IATA by developing case studies, which constitute examples of predictions that are fit for regulatory use. By putting forward these case studies, GOLIATH will actively contribute to the common understanding of using novel methodologies and the generation of considerations/guidance stemming from these case studies [116]. This work will also help to identify areas for developing further guidance on IATA and will be integral to the recommendations to be made to the relevant OECD and European Commission expert groups, regulators and other stakeholders, regarding the selection of appropriate in vitro assays for the future design of a Metabolic Disruption IATA. Uncertainty analysis approaches will be utilized to ensure that uncertainties are identified and transparently documented. In addition, we will identify how the GOLIATH (pre)validated tests can be integrated into the OECD Guidance Document 150, and the European Union EFSA and ECHA Guidance for the identification of endocrine disruptors in the context of Regulations (EU) No 528/2012 and (EC) No 1107/2009 [117].

With a consortium comprised of world-leading experts, GOLIATH will be pivotal in the development of an internationally harmonized strategy for testing and assessing MDCs, with the ultimate aim of protecting human health, slowing the worldwide rise in metabolic disorders that have reached “Goliathan” proportions.

## Figures and Tables

**Figure 1 ijms-21-03480-f001:**
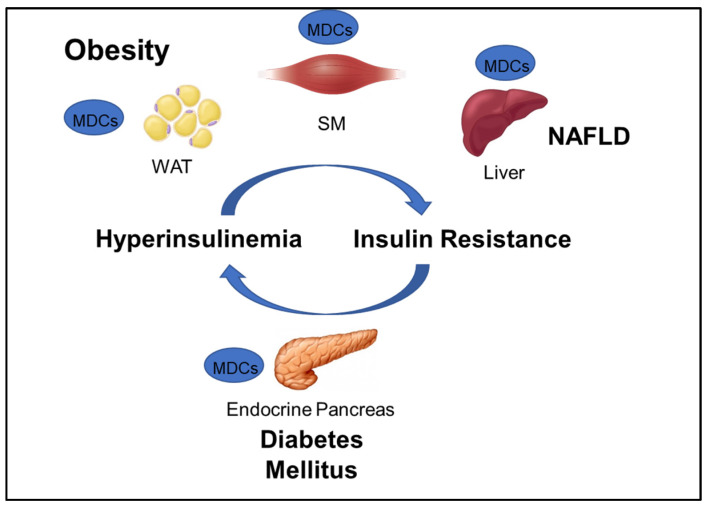
Interconnection between obesity and major metabolic disorders and the corresponding tissues that can be targeted by metabolic disrupting chemicals (MDCs). NAFLD: non-alcoholic fatty liver disease; WAT: white adipose tissue, SM: skeletal muscle.

**Figure 2 ijms-21-03480-f002:**
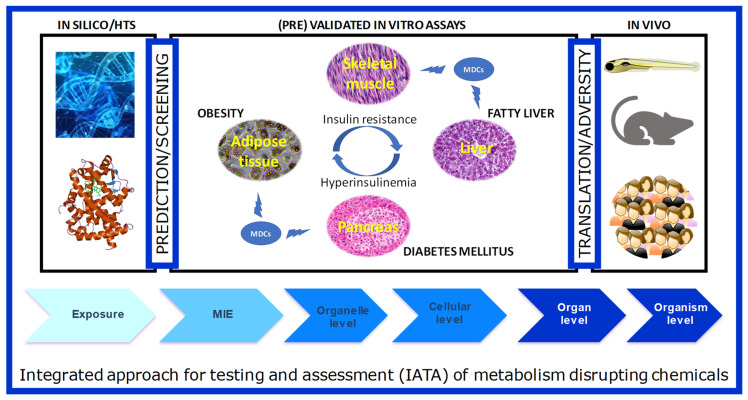
The schematic framework of test method development in the GOLIATH project in an Adverse Outcome Framework. Ultimately, the tests developed and pre-validated within GOLIATH will be integrated to form an IATA for metabolism disrupting chemicals.

**Table 1 ijms-21-03480-t001:** Classes of MDCs and the initial set of model compounds that will be tested in GOLIATH, putative human nuclear receptor (NR)-mediated mechanism of action (MoA) and weight of evidence from animal and human studies as well as the availability of data on human exposure from biomonitoring studies.

Chemical	Identifiers (CAS; ChEBI; Inchikey)	Structure	MoA	Animal	Human	Biomon. Data
Bisphenol A (BPA)	80-05-7; 33216; IISBACLAFKSPIT-UHFFFAOYSA-N	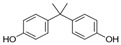	hERα hERβ hPXR hCAR	++	+	+
Tributyltin Chloride (TBT)	1461-22-9; 79734; GCTFWCDSFPMHHS-UHFFFAOYSA-M	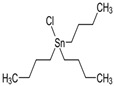	hRXR hPPARγ	+++	-	-
Perfluorooctanoic acid (PFOA)	3825-26-1; 35549; SNGREZUHAYWORS-UHFFFAOYSA-N	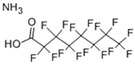	hPXR hPPARγ	++	+	++
Triphenylphosphate (TPP)	115-86-6; 35033; XZZNDPSIHUTMOC-UHFFFAOYSA-N	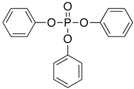	hERα hERβ hPXR hCAR hPPARγ	++	±	+
Dichlorodiphenyldichloroethylene (pp’-DDE)	72-55-9; 16598; UCNVFOCBFJOQAL-UHFFFAOYSA-N	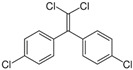	hERα hERβ hPXR hCAR	++	++	+++
Triclosan (TCS)	3380-34-5; 164200; XEFQLINVKFYRCS-UHFFFAOYSA-N	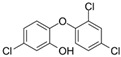	hPXR hPPARγ	+	±	++

Weight of evidence key: - = absence of evidence; ± = equivocal evidence; + = some evidence; ++ = moderate evidence; +++ = strong evidence.

**Table 2 ijms-21-03480-t002:** Effects of MDCs on glucose consumption and insulin sensitivity in in vitro adipocyte models.

Chemical	Cell Type	Effects	Concentrations at which Effects Observed	References
**BPA**	3T3-L1 adipocytes	- Induction of proinflammatory cytokines - Increased basal glucose consumption and reduced insulin sensitivity	1 & 100 nM	[87,88]
**BPA**	3T3-F442A adipocytes	- Increased basal and insulin-stimulated glucose uptake - Increased GLUT4 expression	100 μM 1 & 100 μM	[89]
**TBT**	Differentiating 3T3-L1 adipocytes	- Increased basal and insulin-stimulated glucose uptake - No effect on GLUT4 mRNA expression	50 nM	[90]
**TBT**	Differentiating mMSC adipocytes	- Decreased adiponectin expression - No effect on basal or insulin-stimulated glucose uptake, and pAkt	100 nM	[81]
**PFOA**	3T3-L1 adipocytes	- Increased GLUT4 mRNA expression	200 μM	[91]
**ppDDE**	3T3-L1 adipocytes	- Increased adiponectin - No effect on glucose uptake	2 & 20 μM	[92,93]
**TPP**	3T3-L1 adipocytes	- Increased basal glucose uptake - Increased insulin-stimulated glucose uptake over long time	0.1–50 μM 50 μM	[79]

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
