# Peer review of "The GOLIATH Project: Towards an Internationally Harmonised Approach for Testing Metabolism Disrupting Compounds"

_ijms, 2020, doi:10.3390/ijms21103480_

Round 1

Reviewer 1 Report

The goal of the manuscript is to present the GOLIATH project funded by the EU. The manuscript is only descriptive and it does not include results.

The project started on January 1, 2019 (https://cordis.europa.eu/project/id/825489/fr) under these conditions, it is late to submit such a presentation manuscript especially when specific articles have already been published (https://goliath.wp.hum.uu.nl/publications/).

The interest to publish such a manuscript is very limited.

In addition, these lengthy descriptions are often confusing. It lacks a timetable with what would be obtained at each step.

Miscellaneous

- Page 6, first line of the second paragraph of “In silico predictive models”, print “Schneider et al  [22],” in place of “Schneider et al (2019) [22],”

- Page 10, first line: Print “Lo et al. [86]” in place of “Lo et al. (2013) [86]”

- Page 12, second paragraph, line 5: Print “Jacobs et al [115]” in place of “Jacobs et al 2016 [115]”

Author Response

We appreciate the comments of Reviewer 1.

This manuscript does indeed have the goal to present the GOLIATH project and its main goals, as well as provide a concise review of the literate.  As a 'project report' it is indeed descriptive and does not include results yet, because no results have been obtained yet.  Reviewer 1 argues that the interest to publish such a manuscript our limited. However, I would respond by saying that similar project reports are highly cited, and that it is important to raise awareness of this project, as well as the other 7 projects in the EURION cluster, because they are unique in the world. As the project is in early days, we are also open to feedback from others as well as collaboration with others. 

Reviewer 1 states that as the project started on January 1, 2019, it is late to submit such a presentation manuscript especially when specific articles have already been published (see GOLIATH website https://goliath.wp.hum.uu.nl/publications/). However, articles presenting results of the GOLIATH project have not yet be published. The two articles referred to on our website are review manuscripts. These articles have been referred to in the main text of the manuscript (reference 22 and 29). 

Reviewer 1 states that the lengthy descriptions are confusing and lack a timetable with what would be obtained at each step.  We have addressed this point by adding the timing of each method to be developed to the text. 

Miscellaneous - all of the points below have been corrected

- Page 6, first line of the second paragraph of “In silico predictive models”, print “Schneider et al  [22],” in place of “Schneider et al (2019) [22],”

- Page 10, first line: Print “Lo et al. [86]” in place of “Lo et al. (2013) [86]”

- Page 12, second paragraph, line 5: Print “Jacobs et al [115]” in place of “Jacobs et al 2016 [115]”

Reviewer 2 Report

The report “The GOLIATH Project: Towards An Internationally Harmonised Approach for Testing Metabolism Disrupting Compounds” will provide key information on MDC-related mode of action by incorporating multi-omic analyses, and translating results from in silico, in vitro and in vivo models and assays to adverse metabolic health outcomes in humans at real life exposures.

The report is well written, and I will recommend it for publication in IJMS after authors respond to the following minor comments

Comments:

  • Authors should include line numbers.
  • In the abstract, the sentence “The purpose of this project report is to introduce the European ‘GOLIATH’ project, a new research project which addresses one of the most urgent regulatory needs in the testing of endocrine disrupting chemicals (EDCs), namely the lack of methods for testing EDCs that disrupt metabolism and metabolic functions – chemicals collectively referred to as ‘metabolism disrupting compounds’ (MDCs)” is too long. Please break this down into different sentences.
  • Is there any particular reason for selecting these chemicals; BPA, TBT, and so on? Page 4
  • Why did authors select a cohort centered on asthma and mite allergy? Page 11

Author Response

We thank Reviewer 2 for the useful comments. I will respond to the comments one by one: 

  • Authors should include line numbers.

The manuscript is formatted according to IJMS guidelines. No line numbers are required.

  • In the abstract, the sentence “The purpose of this project report is to introduce the European ‘GOLIATH’ project, a new research project which addresses one of the most urgent regulatory needs in the testing of endocrine disrupting chemicals (EDCs), namely the lack of methods for testing EDCs that disrupt metabolism and metabolic functions – chemicals collectively referred to as ‘metabolism disrupting compounds’ (MDCs)” is too long. Please break this down into different sentences.

This sentence has been broken down into 2 sentences. 

  • Is there any particular reason for selecting these chemicals; BPA, TBT, and so on? Page 4

An explanation for the selection of these chemicals has been added to page 4:

'This initial set of model test chemicals was selected because they have a wide range of mechanisms of action (MoA), and there is existing data from animal and/or human studies which indicate metabolism disrupting effects following exposure. In addition, all of these six chemicals have been detected in humans in biomonitoring studies.'

  • Why did authors select a cohort centered on asthma and mite allergy? Page 11

The  “Prevention and Incidence of Asthma and Mite Allergy” PIAMA cohort was originally established in 1996 to study asthma and mite allergy. However, as stated on page 11, this cohort has extensive information on  lifestyle and environmental exposures as well as information on metabolically relevant molecular and physical outcome parameters at multiple ages in childhood and adolescence. It is called the PIAMA cohort, but it is well designed to examine associations between exposures to MDCs and metabolic outcomes. In addition, as mentioned on page 11, inclusion of the cohort allows us to the important opportunity to access untargeted high-resolution mass spectrometry data (exposome scan) to screen and quantify MDCs and hundreds of other environmental chemicals in addition to endogenous metabolites 

Reviewer 3 Report

excellent summary of an ongoing consortia project with solid partners and high level projects. My advice is to team up with groups working with relevant in vivo mouse models as zebra fish may not be complex enough to reveal targets/mode of action.

Author Response

We appreciate the positive feedback from Reviewer 3.  In response to Reviewer 3's advice to ' team up with groups working with relevant in vivo mouse models as zebra fish may not be complex enough to reveal targets/mode of action', indeed, though we will not perform in vivo mouse studies ourselves, we will work together with other projects within the EURION consortium, as well as outside the cluster, to have access to mouse data. 

We have added a statement to page 11 to this extent:

'Note that though no new in vivo studies will be conducted within this project, partners within the GOLIATH project will team up with other partners within and outside the EURION cluster who are performing rodent studies to enable translational and cross-species comparative analyses.'